# Long-term in vitro 2D-culture of *SDHB* and *SDHD*-related human paragangliomas and pheochromocytomas

Jean-Pierre Bayley[1]*, Heggert G. Rebel[1], Kimberly Scheurwater[1], Dominique Duesman[1], Juan Zhang[1], Francesca Schiavi[2], Esther Korpershoek[3], Jeroen C. Jansen[4], Abbey Schepers[5], Peter Devilee[1,6]

1 Department of Human Genetics, Leiden University Medical Centre, Leiden, The Netherlands, 2 Veneto Institute of Oncology, University of Padova, Padova, Italy, 3 Department of Pathology, Josephine Nefkens Institute, Erasmus MC, Rotterdam, The Netherlands, 4 Department of Otorhinolaryngology/Head and Neck Surgery, Leiden University Medical Center, Leiden, The Netherlands, 5 Department of Internal Medicine, Leiden University Medical Centre, Leiden, The Netherlands, 6 Department of Pathology, Leiden University Medical Center, Leiden, The Netherlands

* j.p.l.bayley@lumc.nl

**Data Availability Statement:** All relevant data are within the paper and its Supporting Information files.

## Abstract

The neuroendocrine tumours paraganglioma and pheochromocytoma (PPGLs) are commonly associated with succinate dehydrogenase (SDH) gene variants, but no human SDH-related PPGL-derived cell line has been developed to date. The aim of this study was to systematically explore practical issues related to the classical 2D-culture of SDH-related human paragangliomas and pheochromocytomas, with the ultimate goal of identifying a viable tumour-derived cell line. PPGL tumour tissue/cells (chromaffin cells) were cultured in a variety of media formulations and supplements. Tumour explants and dissociated primary tumour cells were cultured and stained with a range of antibodies to identify markers suitable for use in human PPGL culture. We cultured 62 PPGLs, including tumours with confirmed *SDHB*, *SDHC* and *SDHD* variants, as well as several metastatic tumours. Testing a wide range of basic cell culture media and supplements, we noted a marked decline in chromaffin cell numbers over a 4–8 week period but the persistence of small numbers of synaptophysin/tyrosine hydroxylase-positive chromaffin cells for up to 99 weeks. In cell culture, immunohistochemical staining for chromogranin A and neuron-specific enolase was generally negative in chromaffin cells, while staining for synaptophysin and tyrosine hydroxylase was generally positive. GFAP showed the most consistent staining of type II sustentacular cells. Of the media tested, low serum or serum-free media best sustained relative chromaffin cell numbers, while lactate enhanced the survival of synaptophysin-positive cells. Synaptophysin-positive PPGL tumour cells persist in culture for long periods but show little evidence of proliferation. Synaptophysin was the most consistent cell marker for chromaffin cells and GFAP the best marker for sustentacular cells in human PPGL cultures.

**Funding:** This work was supported by the Paradifference Foundation (no grant number) (http://www.paradifference.org/) and the Dutch Cancer Society (Grant 2011-5025) (https://www.kwf.nl). The funders had no role in study design, data collection and analysis, decision to publish, or preparation of the manuscript.

**Competing interests:** The authors have declared that no competing interests exist.

## Introduction

Hereditary pheochromocytoma-sympathetic paragangliomas (PPGL) and parasympathetic head & neck paragangliomas (HNPGL) are two distinct neuroendocrine tumour types both primarily caused by variants in succinate dehydrogenase (SDH) subunit genes, all of which are components of a single protein complex of the respiratory chain referred to as Complex II. Sympathetic paragangliomas are more often aggressive and metastatic, whereas parasympathetic head & neck paragangliomas cause considerable local morbidity. For the sake of brevity, we refer to these tumours collectively as PPGLs, except where specific tumour types are discussed.

Around 40% of all PPGLs have a known genetic cause and germline or somatic variants have been identified in 19 putative PPGL genes to date [1, 2]. In addition to the well-known cancer syndromes caused by the *RET*, *VHL* or *NF1* genes that include pheochromocytomas, and the SDH genes (*SDHA*, *SDHB*, *SDHC*, *SDHD*, *SDHAF2*) that are primarily associated with PPGL, recently identified suspected or confirmed PPGL-associated genes include *HRAS*, *EPAS1* [HIF2A], *FH*, *MDH2*, *IDH1*, *IDH2*, *DLST*, *SLC25A11*, *GOT2*, *SUCGL2*, *TMEM127* and *MAX*.

Benign paragangliomas often maintain the typical morphology of a normal paraganglion [3] and consist of two principal cell types, the 'chromaffin' or 'chief' cells (also known as 'type I' cells), which are the neoplastic component [4–8], and non-neoplastic sustentacular cells ('type II' cells), a cell type thought to have a tissue-supportive function in normal paraganglia. Strictly speaking, 'chromaffin' is an inaccurate designation for paraganglioma cells of the head and neck as the traditional potassium dichromate chromaffin reaction, based on the oxidation of stored catecholamines, generally fails to produce a noticeable colour shift in these cells due to relatively low catecholamine production. Nonetheless, the term is now commonly used to refer to all paraganglioma-pheochromocytoma tumour cells.

Chromaffin cells have a large, eccentric cell nucleus relative to the sparse pale cytoplasm and are usually found arranged in rounded cell nests traditionally referred to as 'zellballen'. Sustentacular cells (type II cell) have a flattened, elongated nucleus and a thin, flattened cytoplasm. Together a number of these cells may completely envelope a 'nest' of chromaffin cells [9]. These two cell types constitute the bulk of the central 'zellbal' structure of the paraganglion, but individual cell nests and especially groups of cell nests are often embedded in dense stroma and vasculature. Even large tumours often maintain the characteristic morphology of the normal paraganglion, suggesting that neoplastic cells maintain control of normal tissue plasticity and coordinate the growth of sustentacular and other non-neoplastic cells such as fibroblasts and endothelial cells. Metastatic tumours often lose this typical architecture and display a less differentiated morphology and far fewer sustentacular cells [10].

Paragangliomas and pheochromocytomas typically exhibit low growth rates, with estimated doubling times of 4 to 7 years [11–13]. Even paragangliomas and pheochromocytomas associated with succinate dehydrogenase subunit B (*SDHB*) variants, which may result in aggressive metastatic disease and fatal outcomes, show an overall 5-year survival of 76%-85% [12, 14]. In benign tumours, Ki-67 immunoreactivity as a measure of proliferation often stains less than 1% of all cells. Interestingly, some primary tumours show higher rates of proliferation (16–29% positive cells), with metastases showing levels similar to the primary tumour [15].

Sympathetic pheochromocytomas-paragangliomas are associated with a wider range of genes in both humans and animal models compared to parasympathetic paragangliomas [2, 16]. Culturing pheochromocytomas from animal tumours led to the development of the rat PC12 cell line [17], which carries a Max gene variant, and the MPC [18] and derived MTT [19] mouse cell lines that originated from the *Nf1* knockout mouse, as well as the more recent RS0 cell line from rats lacking SDHB [20]. In contrast to the availability and interest shown in

pheochromocytoma, paraganglioma cell culture has received much less attention. In 1962 Costero and Chevez [21] described certain morphological features of two carotid body tumour (CBT) cultures. In 1976 Gullotta and Helpap [22] cultured three extra-adrenal paragangliomas, including a carotid body tumour. Very little cell proliferation was observed. The most recent report that focused exclusively on the culture of paraganglioma cells was published in 1981 [23]. To the best of our knowledge, no description of paraganglioma culture has been published in the intervening four decades. Although further work has likely been undertaken, the outcome of these studies has remained unpublished, frustrating efforts to distinguish useful approaches. The need for cell models of PPGL is becoming more urgent as basic research transitions from the study of PPGL genetics to the study of the molecular mechanisms driving tumourigenesis and the clinical vulnerabilities of PPGLs. This transition will require genuinely relevant models if further progress is not to be constrained.

Our primary goal in this project was to culture a large number of primary paragangliomas and pheochromocytomas in search of a tumour that had already undergone the genetic changes necessary to allow growth in culture. This strategy was based on the fact that most tumour cultures, regardless of type, do not give rise to cell lines. Even amongst tumour types that readily produce cell lines, such as colon tumours, significant effort is required to identify the small proportion of tumours that yield cell lines. A second goal was to test a variety of basic cell culture media, as well as general or specific supplements, to identify conditions that might promote the survival or proliferation of chromaffin cells in vitro. A third goal was to identify the various cell types found in tumour cultures.

We describe the long-term culture of chiefly head and neck paragangliomas using primary tumour tissue from anonymous donors who carried succinate dehydrogenase subunit B (*SDHB*), subunit C (*SDHC*), subunit D (*SDHD*) or undefined gene variants. Sixty-two tumours were cultured in total, including adrenal pheochromocytomas and extra-adrenal paragangliomas. We evaluated a wide variety of culture media and supplements and we determined which of the specific protein cell markers used in immunohistochemical analysis are suitable for use in the identification of specific cell types in human HNPGL/PPGL cultures. While we identified useful analytical approaches and some basic conditions of culture, our primary of aim of developing a viable PPGL cell line was not achieved.

## Materials and methods

### Surgical material

Tumour tissue was obtained from irreversibly anonymized donors as surgical waste material that would have otherwise been incinerated. This study was therefore issued with a waiver by the Leiden University Medical Ethics Committee (CME protocol P12.082). The pathologist provided information on the location of the tumour and the gene involved, if known. Material was collected in serum-free medium (DMEM/F-12 with GlutaMAX, 31331–028, GIBCO Thermo Fisher Scientific) and retrieved from the pathology department as soon as possible after removal but generally within two hours of surgery. Some tissue was obtained from outside sources (NL/Italy) and was transported in serum-free medium at room temperature. Tumour tissue was weighed upon receipt and processed in a sterile flow cabinet. Two sterile disposable scalpels were used to mince tissue while bathed in calcium/magnesium-free Hanks' Balanced Salt Solution (HBSS, Cat. No. 14175–053, GIBCO® Thermo Fisher Scientific) in a sterile Petri dish. Two blocks of tissue of approximately 3x3 mm were first cut, one for fixation in buffered 10% formalin overnight and the other for freezing in Tissue-Tek ™ O.C.T. (Optimal Cutting Temperature compound, Cat. No. 14-373-65, Thermo Fisher Scientific) to allow frozen tissue sections to be prepared. Following overnight fixation, formalin-fixed tissue was

washed once in 70% EtOH and stored in 70% EtOH prior to processing and paraffin-embedding for histochemistry. The tissue placed in O.C.T. was attached to a small cork support, inserted into a sterile 1.5 ml ampoule, snap frozen in liquid nitrogen and immediately transferred to liquid nitrogen storage.

The remaining tumour tissue was minced and washed with calcium/magnesium-free HBSS. During the washing step, a sample was examined under the microscope to check whether a significant number of free tumour cells were visible in the wash solution. Although rarely the case, if a large number of tumour cells were already visible at this point, the fluid was collected, subjected to slow centrifugation at 20 g for 5 minutes to help preserve cell viability and the pelleted cells resuspended in a small volume of calcium/magnesium-free HBSS.

Of the available washed and minced tissue around 200 ul by volume was transferred to a sterile 1.5 ml ampoule and flash frozen in liquid nitrogen for later DNA/RNA/protein extraction. Three equal volumes of approximately 200 ul tissue were then divided over 3 sterile 1.5 ml ampoules containing 1 ml of cooled freezing medium for cryopreservation of viable tissue (Recovery ™ Cell Culture Freezing Medium, Cat. No. 12648–010, GIBCO® Thermo Fisher Scientific), gently mixed and placed in a pre-cooled isopropanol container before transfer to a -80˚C freezer for overnight storage. Samples were transferred to liquid nitrogen storage the next day. The remaining chopped tissue was initially divided into two parts, with one half used for collagenase/dispase dissociation and the other half for direct culture as tissue explants in a variety of cell media (S1 Table, Tabs Media, Supplements). When insufficient tissue was available, all remaining tissue was treated with collagenase/dispase and all later experiments focused exclusively on enzyme-dissociated tissue.

## Processing of tissue

Both explants and tissue destined for further digestion were first minced in a Petri dish as described above. Tumour tissue was directly cultured as small tumour fragments (explants) of approximately 1 mm$^3$ or was first digested with collagenase-dispase only (Sigma Aldrich-Roche, Cat. No. 10 269 638 001) as a relatively mild enzyme treatment or by addition of collagenase B (Sigma Aldrich-Roche Cat. No. 11 088 815 001), a more aggressive treatment for tumours resistant to collagenase-dispase digestion. Enzyme solutions were reconstituted in Hank's Balanced Salt Solution (HBSS) with $Ca^{2+}/Mg^{2+}$ (GIBCO Thermo Fisher Scientific, Cat. No. 14065–049) and sterilized using a 0.20/0.22 SFCA filter. Initial digestion was for 1 hour in a water bath at 37˚C, with hand mixing every 15 minutes, after which a sample was assessed under the microscope to judge the extent of digestion, in addition to a visual assessment of the sample tube. Digested tissue tends to increase in volume in the early phases of a successful digestion. Longer digestion times were used where necessary, including overnight digestion at 4˚C in the case of very tough tissues. Digestions requiring collagenase B were kept as brief as possible and rarely lasted more than 2 hours as longer digestion times were visibly detrimental to cell survival. When digestion was complete based on visual and microscopic examination of a sample, digested tissue was centrifuged gently at 20 g. Digested tissue was then washed in 5 ml Ca/Mg -/- HBSS, re-centrifuged at 20 g and resuspended in the appropriate culture medium and cultured in a 6-well or 12-well plate in a 37˚C/5% $CO_2$ incubator. As supernatants from the wash steps may contain substantial numbers of tumour cells, wash steps were pooled and centrifuged at progressively increasing g to recover as many tumour cells as possible.

## Culture media

Depending on the volume of tumour tissue available, the following types of culture medium were tested*:

- Serum-free medium: DMEM + DMEM/F-12 (ratio 3: 1) + supplements [24]

- 1% FBS medium: DMEM/F-12 + 1% FBS + ITS-A

- 5% FBS medium: DMEM/F-12 + 5% FBS + ITS-A

- IZAL medium: 60% DMEM (Low Glucose) + 40% MCDB-201 + supplements [25]

- PC-12 medium: RPMI1640 + 10% horse serum + 5% FBS [17]

*A detailed overview of the composition of the different culture media can be found in S1 Table.

Primary tumour cell cultures, whether digested cells or explants, were initially cultured in Greiner Bio-One CellStar Polystyrene 6-well or 12-well cell culture plates and were only transferred to 50 ml Greiner Bio-One Cellstar cell culture flasks when a well was approaching 100% confluency. All liquids (medium, PBS, etc.) were warmed to 37°C in a waterbath prior to use. The culture flask was rinsed briefly with 1x PBS, after which 0.5 ml trypsin-EDTA was added and incubated for around 2 minutes at 37°C until all cells detached. Five ml of medium was then added and the cells were transferred to a sterile 50 ml culture flask and returned to the 37°C incubator. Media were refreshed weekly or bi-weekly. Other cultures were transferred using only vigorous pipetting to detach loosely attached cells, followed by transfer to appropriate medium to adjoining wells in 6 or 12-well plates.

## Immunochemical analysis

We used 8-chamber slides (Cat. No. 154534, Thermo Fisher Scientific) to prepare primary cultures and matching control cell lines for analysis. These slides consist of a 25x75 mm glass microscope slide with an 8-well polystyrene culture chamber surround to contain media. To improve cell adhesion, 8-chamber slides were first coated with poly-D-lysine hydrobromide (1 mg/ml, Cat. No. P7405, Sigma-Aldrich). After a 15 minute incubation, the poly-D-lysine solution was removed and wells were briefly washed three times with MQ, followed by drying in a sterile flow cabinet for one hour. Primary tumour cell cultures were then added, in the appropriate medium, to the chamber slides and cultured for a further 3–4 days to allow reattachment and stabilization of the culture before analysis. Chamber slides were prepared for immunostaining by removal of media, gentle rinsing with PBS (with 0.9 mM $CaCl_2$ and 0.5 mM $MgCl_2$) and drying for 1 hr at RT, prior to fixation with 10% PBS-buffered formalin for 10 minutes at RT. This procedure prevented detachment of cells compared to other methods we explored. As staining controls for many of the analyses, paraffin sections from the matching tumour were also prepared using standard immunohistochemical techniques, consisting of antigen retrieval in 10 mM citrate buffer (pH 6.0) for 5 minutes at 110°C in an autoclave or 10 mM TRIS/1 mM EDTA buffer, pH 9.0, for 10 minutes in a microwave, blocking of endogenous peroxidase with methanol/$H_2O_2$ (95%/5%, v/v), PBS with 5% bovine serum albumin to block aspecific antibody binding, triple washes in PBS/Tween (0.05%), followed by incubation with Bright Vision poly-HRP anti-mouse/rabbit (Immunologic), with DAB as the chromophore (Dako, K3468), and counterstaining with haematoxylin.

## Lactate culture

All lactate concentrations were diluted from a 40 mM stock solution of sodium L-Lactate (L-Lactic acid sodium salt ≥99%, cat. No. 71718, Sigma-Aldrich) sterilized in a flow cabinet using a 0.22 μm filter.

Addition of cells to 8-chamber slides was done by first removing medium from a 25 $cm^2$ culture flask containing the primary tumour cell culture, briefly washing with warmed PBS,

followed by incubation with 0.5 ml trypsin-EDTA until cells became detached. Five ml of 1% FBS culture medium was then added to the culture flask and 100 μl of the cell suspension inoculated into wells of an 8-chamber slide, after which 100 μl of medium including the appropriate lactate concentration (0 mM—1.0 mM—1.5 mM—2.5 mM—3.5 mM—5.0 mM—7.5 mM and 20 mM) diluted in 1% FBS culture medium was added to the wells.

Media were refreshed twice weekly over the four-week experimental period, with replacement of 50% of medium with an equal amount of new medium containing the appropriate concentration of lactate. The 8-chamber slides were cultured in a 37°C/5% $CO_2$ incubator and fixed and stained at 1, 2 and 4 weeks.

## Antibodies

We selected a range of antibodies against cell markers that are known to be expressed by and/or specific for different cell types found in PPGLs when analysing formalin-fixed paraffin-embedded tumours (S1 Table–Antibodies). We selected the following markers for chromaffin (chief/type I) cells: anti-chromogranin A (Cat. No. M0869, Clone DAK-A3, DAKO, 1:1000), anti-synaptophysin (Cat. No. NCL-L-SYNAP-299, Leica Biosystems, clone 27G12, 1:1000). anti-tyrosine hydroxylase (Cat.No. ab112, Abcam, 1:2000), anti-neuron-specific enolase (Cat. No. M0873, Clone BBS / NC / VI-H14, DAKO, 1:2000), anti-GATA3 (nuclear staining; Cat. No. 558686, BD Pharmingen, clone L50-823, 1:1000), the cell surface marker anti-CD56 (NCAM) (Invitrogen MA1-46055, clone ERIC-1, 1:500) and anti-CD56-PE (NCAM) (FACS of chromaffin cells) (Biolegend Cat. No. 362507, clone 5.1H11, 1:250). For the identification of sustentacular (type II) cells we selected the following markers: anti-Glial Fibrillary Acidic Protein (Cat. No. M0761, Clone 6F2, DAKO, 1:100) and anti-S100 (Cat. No. Z0311, DAKO, 1:2000).

To detect other cells types reportedly present in PPGLs, the following markers were used: anti-neurofilament protein (Cat. No. M0762, Clone 2F11, DAKO, 1:200) for the identification of neuronal cells; anti-CD31 (Cat. No. M0823, Clone JC70A, DAKO, 1:20) to identify endothelial cells; anti-cytokeratin (Cat. No. M3515, Clone AE1/AE3, DAKO, 1:300) for the identification of epithelial cells; anti-fibronectin (Cat. No. ab6328, Clone IST-9, Abcam, 1:4000), anti-vimentin (Cat. No. M0740, Clone 3B4, DAKO, 1:1000) and anti-smooth muscle actin (Cat. No. M0851, Clone 1A4, DAKO, 1:100) to identify fibroblasts. Anti-Ki-67 (Cat. No. M7240, Clone MIB-1, DAKO, 1:1000) was used as a marker for proliferating cells, and anti-IgG1 (Cat. No. X0931, clone DAK-G01, DAKO, 1:100) and anti-IgG2b (cat. No X0944, Clone DAK-GO9, DAKO, 1:100) were used as negative isotype controls.

## Control cell lines

Paragangliomas are heterogeneous tumours containing many different cell types. When evaluating the various cell types found in these tumours it is useful to have standard cell lines of known origin and morphology available to serve as positive controls. We chose to use the well-established pheochromocytoma cell line PC-12, the neuroblastoma cell line SH-SY5Y for its representative neuronal characteristics, and the non-transformed human fibroblast cell line VH10hTneo as a representative connective tissue cell (S1 Fig). The PC-12 cell line is derived from a rat adrenal pheochromocytoma and shows morphological and phenotypic similarities to the human chromaffin cells found in paragangliomas [17]. The PC-12 cell line was obtained from the American Type Culture Center (PC-12, Cat. No. CRL-1721, ATCC Manassas, Virginia). The SH-SY5Y cell line, derived from a neuroblastoma bone marrow biopsy, is of neuronal origin and exhibits neuronal cell properties. The SH-SY5Y cell line was obtained from the American Type Culture Center (SH-SY5Y, Cat. No. CRL-2266, ATCC Manassas, Virginia).

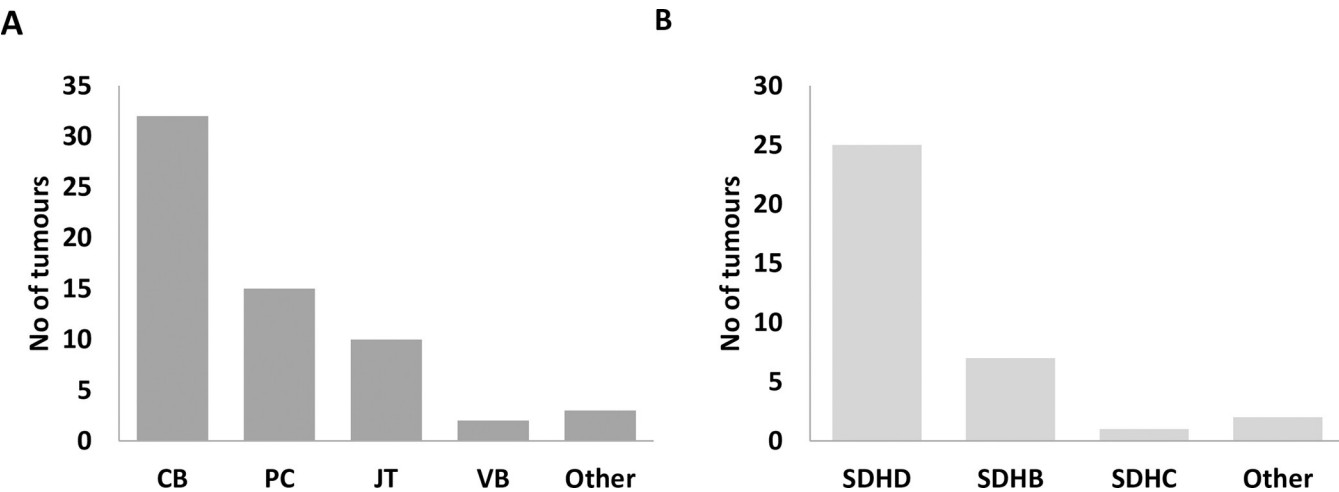

**Fig 1.** Tissue of origin (panel A) and gene variant (panel B) (if known) of the tumours cultured in this study. CB, carotid body tumour; PPGL, pheochromocytoma/extra-adrenal paraganglioma; JT, jugulotympanic tumour; VB, vagal body tumour.

The VH10hTneo cell line is a fibroblast cell line obtained from human foreskin and immortalized using human Telomerase Reverse Transcriptase subunit (hTERT) [26]. Abundant stroma is characteristic of paragangliomas and component cells often quickly dominate paraganglioma tumour cultures. The VH10hTneo cell line was obtained from Ms. Binie Klein (Department of Human Genetics, LUMC, Leiden, NL).

## Results

### Culturing PPGLs

In total, 62 primary paragangliomas and pheochromocytomas were cultured (Fig 1, Table 1). These tumours were mainly obtained from patients operated at Leiden University Medical Center, but some (n = 11) were sourced from other centres in the Netherlands or Italy. The majority were head and neck paragangliomas (Fig 1A), together with 16 non-head and neck PPGLs, including two *SDHB*-associated metastatic tumours. Of the 35 tumours analysed for genetic variants, 25 carried variants in *SDHD*, 7 in *SDHB* and 1 in *SDHC* (Table 1 and Fig 1B). At least two tumours (Tu23, Tu53) were not linked to SDH variants, as determined by sequencing of the SDH genes or retained expression of SDHB upon SDHB immunohistochemical staining.

**Table 1. Summary of the gene variants found in this study.** Abbreviations: SDHB, C & D, succinate dehydrogenase subunits B, C & D, respectively; HNPGL, head & neck paraganglioma; PPGL, pheochromocytoma-sympathetic paraganglioma; meta, metastatic.

|  | HNPGL | PPGL |
|---|---|---|
| SDHB | 4 | 3 (2 meta) |
| SDHC | 1 | 0 |
| SDHD | 23 | 2 |
| Non-SDHx | 2 | 0 |
| Not tested | 16 | 11 |
| Total | 46 | 16 |

## Culture of tumour explants

We initially focused on the culture of tumour explants. The theoretical advantage of explant culture is that it maintains cell-cell interactions and biochemical support from surrounding cells that may be essential to the survival of tumour cells. This type of support may be particularly necessary to the survival of PPGL tumour cells, as evinced by the co-expansion of diverse non-neoplastic cell types in many PPGL tumours. The dissociation of PPGL tumour tissue thus has the potential drawback of disruption of important cell-cell interactions.

However, analysis of several explant cultures over a period of 1–2 months suggested that this approach has major drawbacks, at least in the media used. We cultured tumour explants, removing and fixing fragments in 10% buffered formalin on set days. Histochemical analysis of FFPE tumour fragments revealed that these relatively intact tissues show a major decline in cellularity over this period, probably due to the limited diffusion potential of oxygen and media into relatively substantial (1-2mm x 1-2mm) tissue fragments (Fig 2 and S2 Fig). Furthermore, immunohistochemical analysis using anti-synaptophysin showed that chromaffin cells decline in number, disappearing almost entirely by around 30–45 days of culture. We therefore concluded that the conditions used support chromaffin cell survival for no more than 30–60 days and do not appear to promote cell proliferation. Nevertheless, the characteristic expansion of many other non-tumour cell types in PPGL tumours may play a role in the survival of tumour cells and the role of these cells perhaps shouldn't be too quickly discounted in efforts to culture human tumour cells in vitro.

## Culture of dissociated tumour tissue

In a second set of tumours we focused on cultures of dissociated tumours, in which some chromaffin cells appear to maintain viability for very long periods (>90 weeks). Tumour tissue was first digested using collagenase-dispase as a relatively mild enzyme treatment or by further addition of collagenase B, a much more aggressive treatment only suitable for tumours resistant to collagenase-dispase digestion. Extra care is required when using collagenase B as overdigestion of tissue results in lower numbers of surviving chromaffin cells.

The requirements for enzyme combinations and length of time needed to achieve tissue digestion varied widely between tumours, an effect likely related to the highly variable fibrous extracellular matrix. In general, soft tumours digest quickly (Fig 3A), while very firm tumours digest slowly (Fig 3B). Digestion times are best determined by visual and microscopic estimation of the degree of digestion of samples in a sterile Petri dish after brief trituration. It is worth noting that paragangliomas rarely digest to an exclusively single-cell level. Most cells remain in clumps of a few dozen to a few hundred cells, with interspersed single cells (Fig 3C & 3D). Enzyme wash solutions can be centrifuged at progressively higher speeds to help pellet single cells and cell clumps, avoiding unnecessary loss of tumour cells.

One prominent characteristic of PPGL cultures was the heterogeneity of the appearance and composition of cultures (Fig 3E–3H and S4 Fig). Certain specific aspects of tumour cultures appeared on several occasions or frequently, or appeared progressively as a culture aged. The gross appearance of cultures older than 4–8 weeks was not generally attributable to chromaffin cells, as these cells tended to decline over this period, eventually representing a small minority of surviving cells. The highly variable morphology is therefore likely attributable to connective tissue cells of various types that are also apparent in the gross tumours. Gross tumours vary in appearance from soft, highly vascular tissue to very dark, highly fibrous tissue. It is unclear whether connective tissue cells in different PPGL tumours are distinct cell types or have simply adapted their morphology to prevailing conditions. For instance, long-term cultures occasionally developed areas that showed roughly circular lattice structures (Fig 3E) that

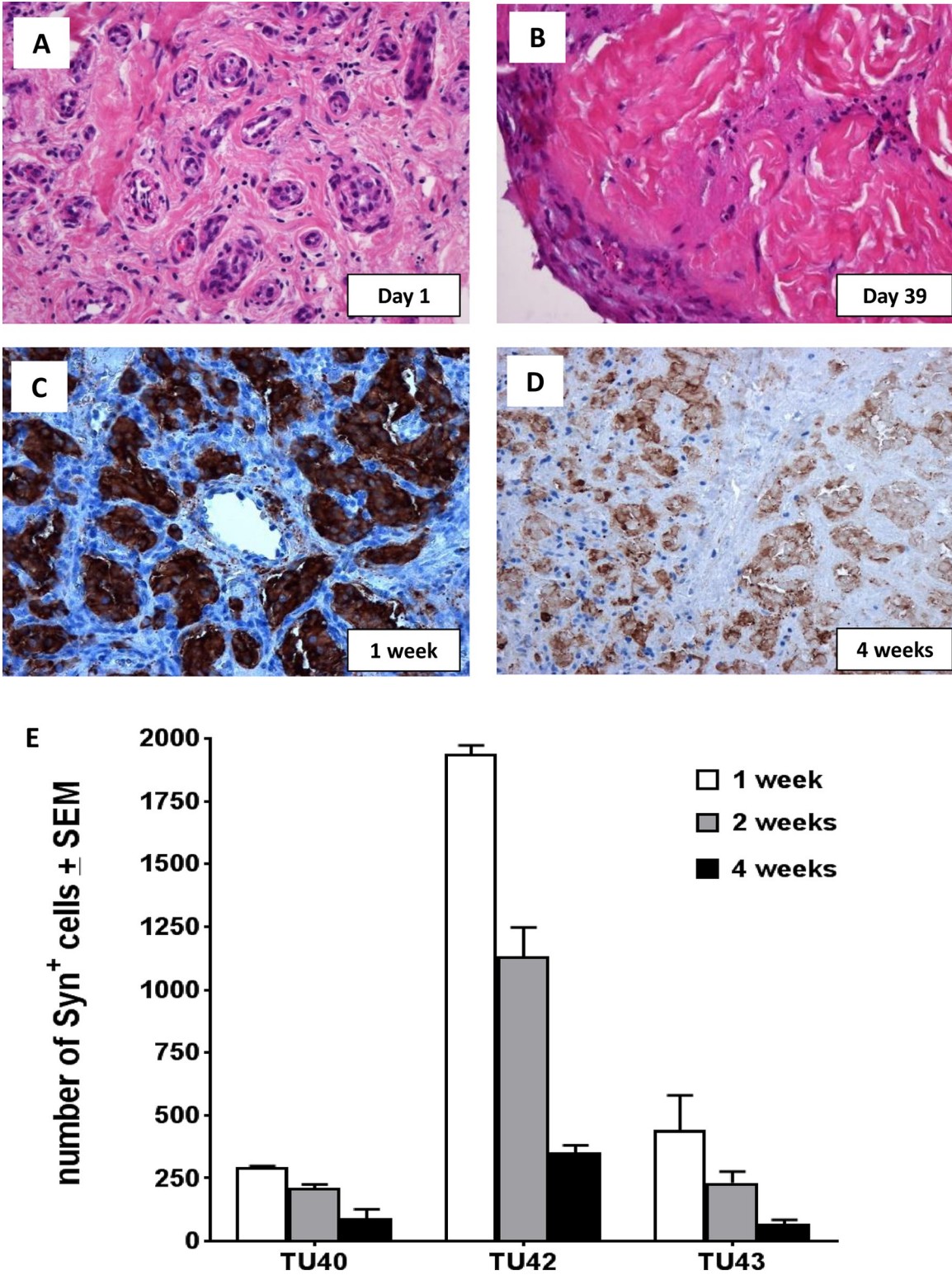

**Fig 2. Tumour explant culture to determine the ongoing cellularity of explants.** Tumour explants (1x1 mm tumour fragments) were cultured for up to 39 days to assess cellularity and remaining chromaffin tumour cells. At day 1, H&E staining of tumour 9 reveals the classic cell nest morphology in a highly stromal background (A, Tu9_CBT_SDHD_H&E_10x obj.). By around day 11 tumour fragments began to show clear signs of declining cellularity and an increase in eosin-avid (pink) acellular proteinaceous material (S2 Fig). By day 39, few cells of chromaffin appearance remained in the tumour fragments (B, Tu9_CBT_SDHD_H&E_10x obj.). Chromaffin cellularity was

determined using immunohistochemistry (C, Tu44_ thoracic PGL_SDHD_Syn_10x obj.) to assess the proportion of synaptophysin-positive cells ('chromaffin' tumour cells) remaining at 1 week (C, Tu44, CBT_SDHD_Syn_10x obj.). At around 4 weeks, (D, Tu44_ thoracic PGL_SDHD_Syn_10x obj.) declining expression of synaptophysin (brown staining) was accompanied by visible shrinkage of cellular tumour areas. Residual chromaffin cell areas can be seen but the lack of nucleic acid (DNA) staining by haematoxylin suggests a loss of cellular integrity. Areas lightly staining for synaptophysin probably consist of cellular debris of chromaffin cells. Quantification of synaptophysin immunohistochemistry in three tumours (E, TU40-CBT, Tu42-PPGL, Tu43-CBT) showed a consistent decline in the expression of this marker protein in explants over a 4-week culture period.

could be due to either sustentacular cells or the connective tissue cells that form the zellballen structures found in intact tumours. As immunocytochemical analysis suggests that sustentacular cells show poor survival in culture (<4 weeks), these structures more likely represent connective tissue cells. This suggests that these cells may have self-organizing capabilities specific to PPGLs and may play a role in or even determine the 'cell nest' structure of these tumours. However, due to the delicacy of structures that tended to develop over a period of several months, we did not have the opportunity to carry out immunocytochemical analysis to unambiguously identify the cell types involved.

## Identifying cell types using immunocytochemistry

Little is known about the survival and proliferation of the various cell types found in PPGL tumours in culture and to the best of our knowledge no publication has comprehensively described the different cell types found in PPGL cultures. We therefore first tested a panel of antibodies that recognize proteins reportedly expressed by various cell types found in PPGL tumours. These antibodies are specific and sensitive when used in the immunohistochemical analysis of FFPE sections (S3 Fig), but their specificity in the immunocytochemical analysis of PPGL cultures has not been previously described. A detailed description of the panel of cell type-specific antibodies tested in PPGL tumour cultures can be found in S1 Table, Antibodies tab.

When used to analyse cell cultures, some markers that are otherwise specific and sensitive in immunohistochemical analysis proved markedly less reliable (Fig 4). One example is chromogranin A, a protein found in secretory vesicles, which is a very reliable marker in tumour sections but was rarely detected in cell cultures. Neuron-specific enolase, an isoenzyme of enolase found in neuronal and neuroendocrine tissues, is also a reliable marker in tumour tissue sections but was often negative in cell culture. The best known and perhaps most commonly used marker for pheochromocytomas is tyrosine hydroxylase, an important enzyme in the synthesis of catecholamines. However, this marker is less suitable for the analysis of HNPGLs, because even primary FFPE tumours may show no detectable expression. Although many tumour cultures were strongly positive for tyrosine hydroxylase, many others were negative, indicating that this marker is also inconsistently expressed in tumour cell cultures. Another commonly used chromaffin cell marker is the synaptic vesicle glycoprotein, synaptophysin. This marker proved the most reliable in cell culture, both in early and late cultures, and showed expression in both undifferentiated and differentiated chromaffin cells. Cells of the same morphology were often positive for either tyrosine hydroxylase or neuron-specific enolase. An overview of these and other immunohistochemical stainings in a representative sample of tumour cultures can be found in Fig 4, S4 Fig and S2 Table.

## Chromaffin cells show long-term survival in PPGL tumour cultures

Having determined that synaptophysin and, to a lesser extent, tyrosine hydroxylase are useful chromaffin cell markers in PPGL tumour cultures, we assessed the expression of these and other markers in a selection of short and long-term cultures. Remarkably, we found that cells

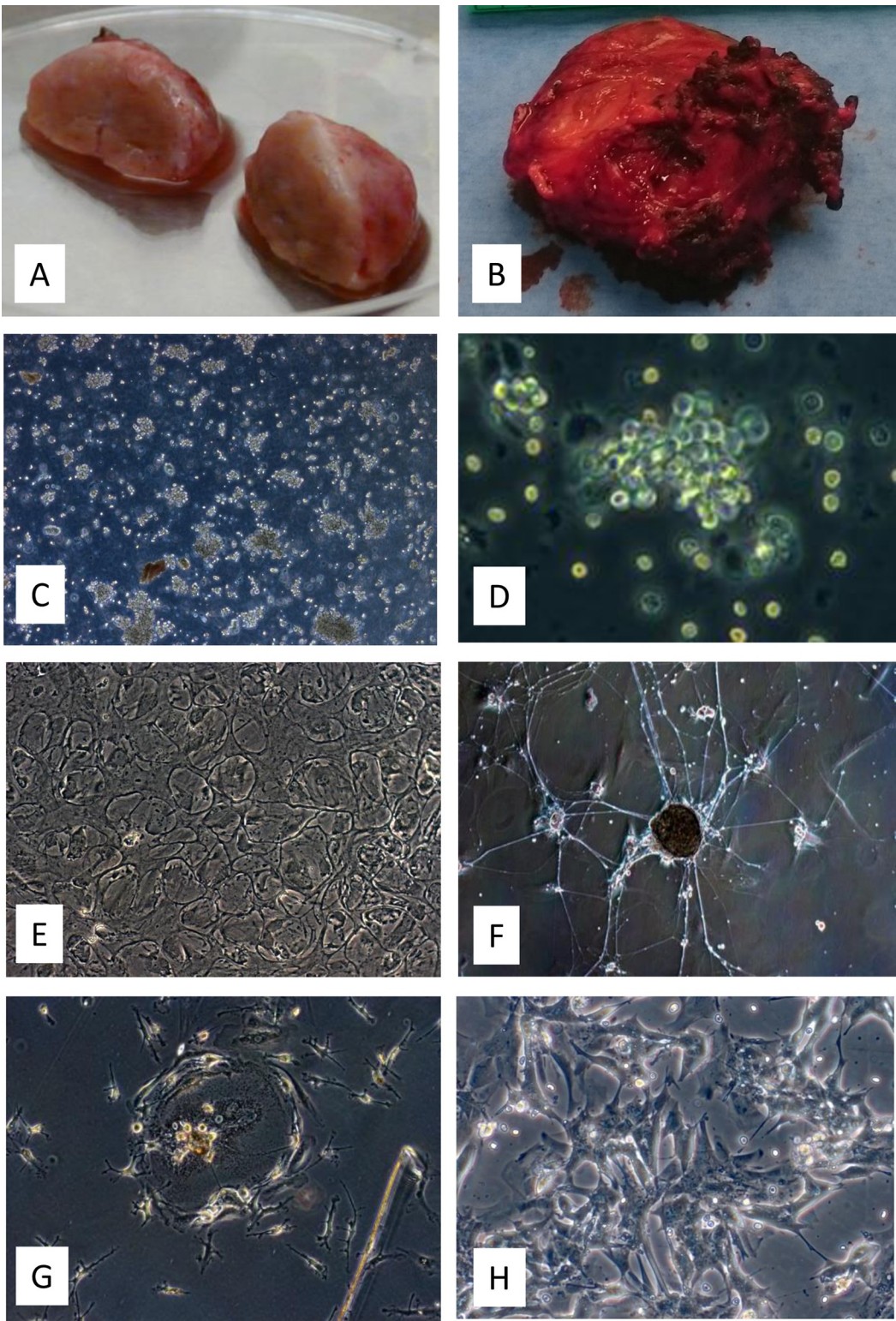

**Fig 3. Surgically excised PPGLs vary considerably in their gross appearance.** Pliant, highly vascular tumours (A) digest readily, whereas firm, dark tumours (B) generally require more rigorous treatment to release tumour cells. Digested tissue features single cells and large and small groups of tumour cells (C, Tu57_10x obj. phase contrast). Tumour-derived cells show an irregular light halo around a darker central area, in contrast to the 'doughnut' appearance of erythrocytes (D, Tu57_40x obj. phase contrast). Long-term cultures occasionally develop areas that show roughly circular lattice structures

(E, Tu11_4 months_5% FBS_10x obj.) that resemble sustentacular cell networks found in tumours but more likely represent connective tissue cells specific to PPGLs, perhaps suggesting that these cells have self-organizing capabilities specific to this tissue. Paraganglioma cell cultures may also develop extensive networks of processes (F, Tu21_4 months_5% FBS_5x obj.), usually but not always after an extended period in culture. These networks can become macroscopically visible in cell culture flasks and usually interconnect discrete cell masses. Cells of tumour culture 23, a paraganglioma without a variant in any commonly-mutated gene, exhibited short, eccentrically branching processes (G, Tu23_20 days_5% FBS_20x obj.), occasionally accumulating around a single, flattened cell. Tumour cells originating from a PPGL bone metastasis showed a semi-differentiated morphology in culture (H, Tu26_6 days_5% FBS_10x obj.), a morphology occasionally seen in other non-metastatic tumours.

positive for tyrosine hydroxylase and/or synaptophysin could still be observed in cultures as late as 99 weeks. Positive cells were generally found in small groups (Fig 4 and S4 Fig) or, in some cases, larger aggregates. It is unclear whether these cell groups and aggregates arise due to limited cell proliferation or represent the long-term survival of an original cell group. It is worth remembering that neither synaptophysin nor tyrosine hydroxylase expression conclusively establishes the identity of a cell as a PPGL chromaffin tumour cell, but in light of the widespread staining in early cultures and the common co-occurrence of positive synaptophysin and tyrosine hydroxylase staining in cells of identical morphology, it is reasonable to assume that these cells are undifferentiated chromaffin tumour cells. In the context of PPGL cell line development the persistence of these cells is encouraging, but it should be noted that immunocytochemical sampling of 33 short and long-term cultures (S3 Table) identified only 4 cultures with substantial numbers (>1%) of synaptophysin- or tyrosine hydroxylase-positive cells, suggesting that even if localized cells were missed when sampling some cultures, a minority of cultures contain persistent chromaffin cells.

## PPGL tumour cultures commonly develop cell agglomerations

A common aspect of paraganglioma culture is the formation of cell agglomerations or masses that appear initially to include many synaptophysin and tyrosine hydroxylase-positive cells but quickly lose these cells or cells lose expression of these markers (Fig 5). The former explanation seems more likely as synaptophysin/tyrosine hydroxylase expression is maintained by other cells of undifferentiated chromaffin morphology for very long periods in culture [S4 Fig, S3 Table]. These cell masses most likely initially form from a mix of chromaffin cells, sustentacular cells and fibroblasts. Within a few weeks the number of chromaffin cells appears to decline and the central cell mass is more often negative for synaptophysin or tyrosine hydroxylase (Fig 5C) and surviving synaptophysin-positive cells congregate at the periphery, presumably where the oxygen/nutrient environment is most favourable. Synaptophysin-positive cells are often seemingly 'shed' from these masses in the initial weeks of culture, visible as individual cells surrounding a cell mass, a phenomenon that can occasionally persist for months in certain cultures. It is unclear whether these cells are actually replicating or are just migrating from within the cell mass, but staining with Ki-67 showed patterns of cell replication that did not appear to overlap with synaptophysin staining patterns (Fig 5G and 5H).

## Cell culture media and cell survival

Growth characteristics and survival of synaptophysin-positive cells in PPGL tumour cultures showed wide variation, some of which appeared attributable to the culture medium. We therefore analysed the proportion of surviving synaptophysin-positive (chromaffin) and GFAP-positive (sustentacular) cells in various media at around 20 weeks of culture (Fig 6). Significant differences between culture media were found for primary cell cultures of tumours 44 (22 weeks) and 46 (20 weeks). The highest percentage of synaptophysin-positive cells was observed

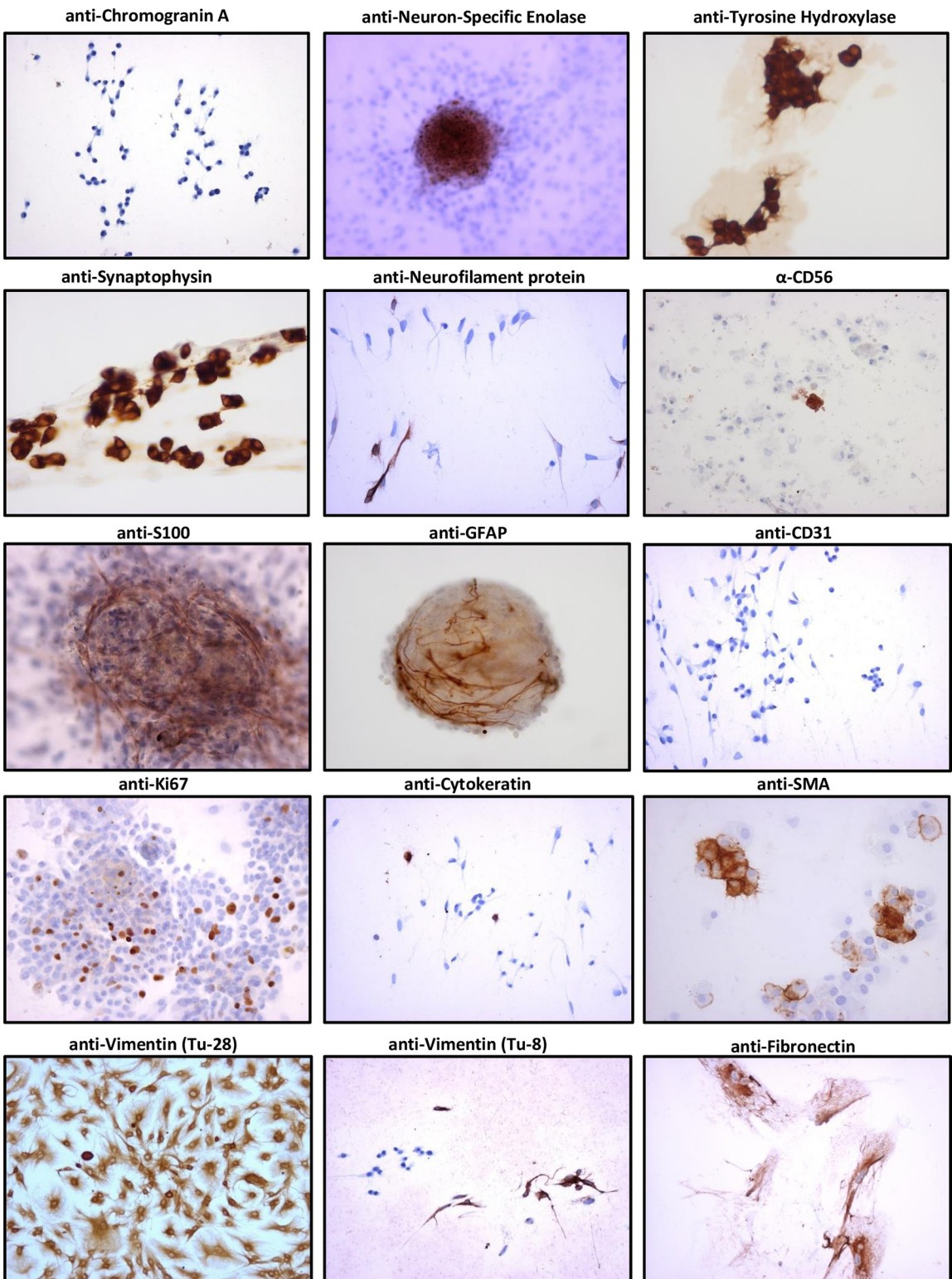

**Fig 4. Summary of immunocytochemical staining of PPGL tumour cultures.** Both chromogranin A (Tu8_Chromogranin A_10x obj.) and neuron-specific enolase (Tu-28_Neuron-specific enolase (NSE)_10x obj.) are generally negative, even in very early cultures. Tyrosine hydroxylase staining (Tu36_Tyrosine hydroxylase (TH)_40x obj.) is often positive, even in long-term cultures, but as many HNPGL cultures are negative for TH at the outset this marker is not generally applicable. In our hands the most broadly reliable marker in PPGL tumour cultures proved to be synaptophysin (Tu18_Syn_40x obj.), as this marker can be found in both early and late cultures.

To assess the survival of neuronal cells in PPGL tumour cell cultures we also stained for neurofilament protein (Tu-7_Neurofilament protein_20x obj.), which is expressed in nerve fibres and the sparse individual neurons seen in PPGL tumours. We found that even at 29 months of culture (Tu7), a few cells of neuronal morphology are still present in culture. CD56 (NCAM) is a cell surface marker for chromaffin cells and is found in most PPGLs, showing the distinct staining pattern of a cell surface protein. Of the few tumour cultures examined (Tu-42_CD56 (ab 123C3) _10x obj.), only sparse cells were positive for CD56. The classic markers for sustentacular cells in PPGLs are S100 and GFAP. S100 tends to show more widespread staining of these cells and is therefore favoured in IHC applications, while GFAP is expressed more sporadically in the same cells. Both marker proteins are expressed in tumour cultures but S100 expression (Tu-40_S100_20x obj.) appears to be short-lived, while GFAP expression (Tu44_thoracic PGL_GFAP_20x obj.) persists somewhat longer, both generally disappearing over the course of 4–8 weeks, along with cells showing sustentacular morphology. CD31 is a marker for endothelial cells and staining of PPGL FFPE tumour sections underscores the extensive vascularity of these tumours. Although few cultures were stained with CD31, those that were appeared uniformly negative (Tu-8_CD31_20x obj.), suggesting that endothelial cells do not persist in culture. Ki-67, a protein expressed in proliferating cells, was occasionally positive in cells of fibroblast morphology (Tu-52_Ki-67_20x obj.). Cytokeratin, a marker for epithelial cells, was occasionally positive (Tu-8_Cytokeratin_10x obj.) in a few cells. Cytoskeleton proteins such as smooth muscle actin (SMA), vimentin and fibronectin are present in most cells and as such are aspecific markers. However, they can usefully illustrate the general morphology of cell cultures. We found SMA (Tu-47_Smooth muscle actin (SMA)_40x obj._cytospin) and vimentin (Tu-28_Vimentin_10x obj.;Tu-8_Vimentin_10x obj.) to be the most useful, with fibronectin only sporadically positive in a few cells (Tu-17 _Fibronectin_20x obj.).

with serum-free medium, a medium developed for the culture of neuroblastomas [24], whereas 'Izal' medium [25] harboured the lowest relative percentage of synaptophysin and GFAP-positive cells. In addition, the percentage of cells positive for Ki-67 was highest in the Izal medium and, qualitatively, this rich medium (S1 Table, Tab media) appeared to best promote fibroblast cell growth across a wide range of tumours, probably contributing to metabolic stress on other cell types. It is important not to overemphasize the benefits of serum-free medium for chromaffin cells, as this medium primarily suppresses the growth of fibroblasts. However, in the short-term it does seem to be the medium of choice if the aim is to obtain a relatively pure tumour cell culture.

## Lactate promotes chromaffin cell survival

Glucose avidity, upregulated glycolysis and the production of L-lactate are all well-known aspects of cancer [27–29]. In recent years it has become clear that L-lactate is not simply a waste product of elevated glucose catabolism; it is also an important component of mitochondrial metabolism in cancer cells [30–32]. We therefore examined the effect of lactate supplementation on chromaffin cell survival in recent primary tumour cultures. Tumour cells from tumours 44 and 46 were introduced into chamber slides together with 1% FBS medium supplemented with lactate at concentrations 0 mM, 1.0 mM, 1.5 mM, 2.5 mM, 3.5 mM, 5.0 mM, 7.5 mM or 20 mM. Medium was refreshed twice weekly by removing 50% and replenishing with an equal amount of new medium with the appropriate concentration of lactate. As seen in Fig 7A, synaptophysin-positive cells from tumour 44 (a thoracic paraganglioma carrying the *SDHD* Dutch founder variant p.Asp92Tyr) were noticeably more persistent in cultures with higher lactate levels. Quantification of positive staining cells at 1, 2 and 4 weeks in Fig 7B illustrates the typical rapid decline in overall numbers of chromaffin cells. Nevertheless, synaptophysin-positive cells persisted significantly longer in culture media containing higher levels of lactate.

## Discussion

### Primary goal

The development of cell lines commonly involves the culture of a large number of tumours in order to identify the small minority that survive in culture [33]. One of our primary goals was therefore to culture as many primary paragangliomas and pheochromocytomas as possible, within the constraints of time and budget, in search of a PPGL tumour that harboured genetic

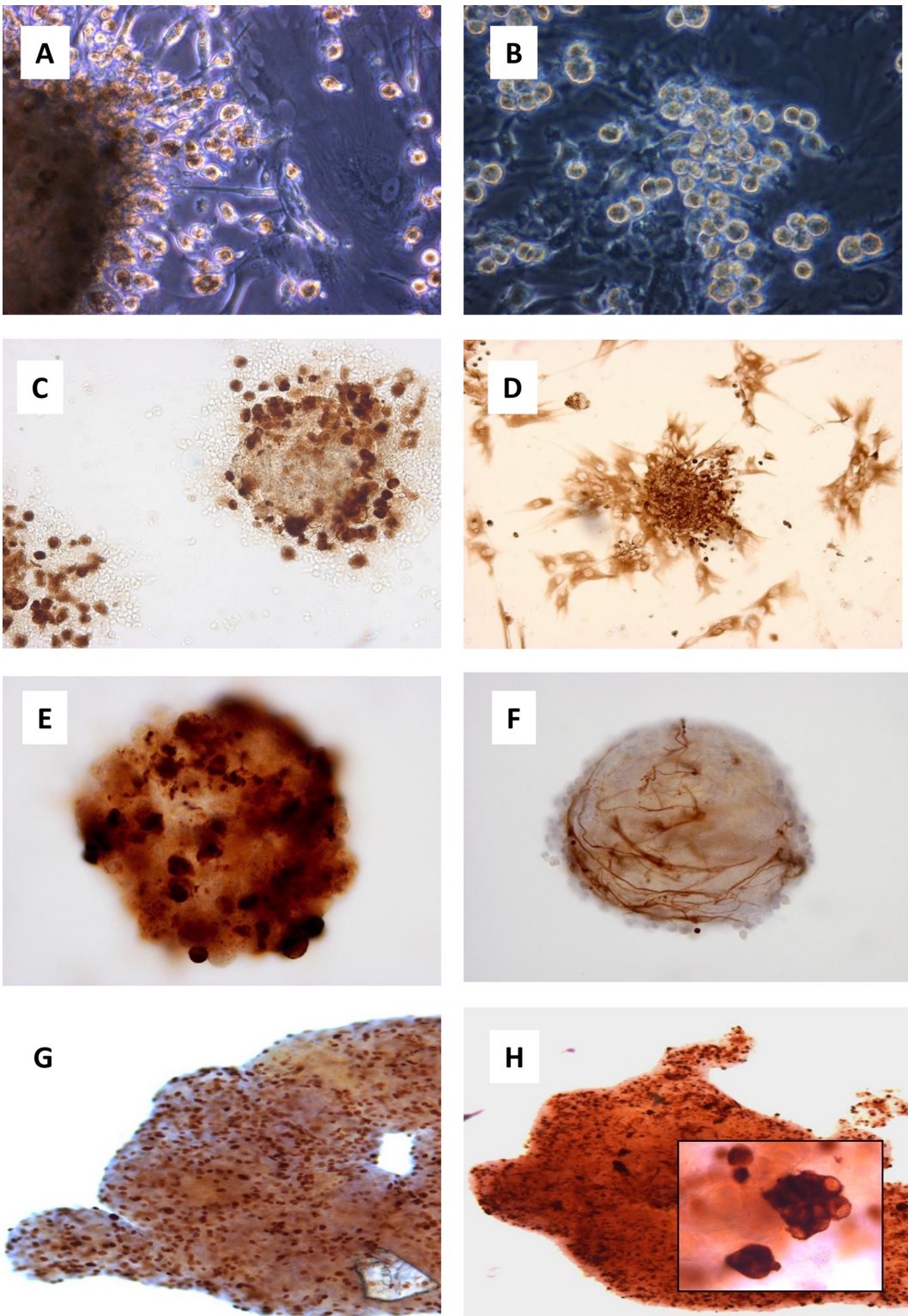

**Fig 5. Large and small cell masses are a common feature of paraganglioma cultures.** Cell masses appear to be anchored by fibroblast-like cells to the plastic substrate, and often 'shed' cells of chromaffin cell morphology during the first few weeks of culture (A, Tu42_35 days_5% FBS_20x obj. phase contrast). 'Shed' cells are often weakly adherent to the plastic or fibroblast underlayer and can be harvested by vigorous pipetting and re-cultured (B, Tu51_58 days _1% FBS_40x obj. phase contrast). These large cell masses consist of numerous synaptophysin-positive cells in the first weeks of culture (C, Tu44_ thoracic

PGL_14 days_1% FBS_Syn_20x obj.). Vimentin antibody staining illustrates the general structure of these cell masses (D; Tu44_14 days_1% FBS_Vim_20x obj.). Cell masses often initially express synaptophysin intensely but gradually lose expression and by 4–6 weeks show synaptophysin-expressing cells only in the outermost layer (E, Tu44_4 weeks_5% FBS_Syn_40x obj.) Cells of sustentacular morphology often associate with cell masses when these cells are present in a culture (F, Tu44_4 weeks_5% FBS_GFAP_ 20x obj.). Cell proliferation is ongoing in cell masses (G, Tu44_4 wks_serum-free med_Ki-67_5x obj.) but does not appear to overlap with surviving synaptophysin-positive cells (H, Tu44_4 wks_serum-free med_Synaptophysin_5x obj.; inset Synaptophysin_40x obj.).

changes compatible with growth in culture. In general, PPGLs show relatively few genetic changes compared to other tumours and little evidence of significant genetic instability, so rapid genetic change in culture ('adaptation to culture') seems unlikely. To allow PPGL tumour cells with proliferative potential sufficient time to show evidence of expansion (a slow process in vivo) we maintained some cultures for up to 2 years. Secondary goals were to identify reliable cell-specific markers in cell culture, media and growth factors that might promote PPGL tumour cell survival and growth in culture, and to study the value of specific culture supplements.

## Foundational study

This study provides a foundation for other researchers wishing to culture human PPGLs. Despite interest in this subject since the 1960's, no detailed description of the appearance and

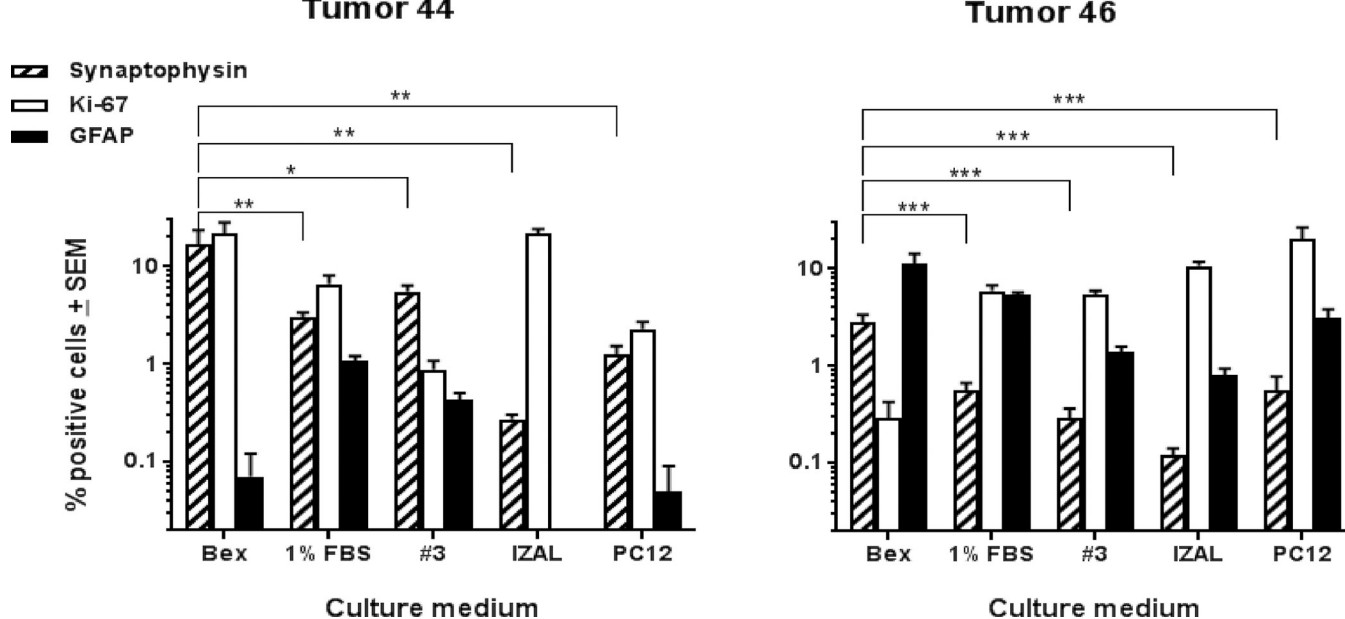

**Fig 6. Comparison of five cell culture media.** Tested on Tu44 (thoracic PPGL, SDHD p.(D92Y)) and Tu46 (carotid body PGL, SDHD p.(L95P)), stained with 3 marker antibodies and scored double blinded in duplo. Percentages of positive cells, with standard error of the mean (SEM), are presented in 10log scale. P values calculated with one-way ANOVA (p < 0.005) and Least Significant Differences (LSD: * p < 0.05, ** p < 0.01, ***p < 0.005). The majority of tumours were cultured in 5 media formulations (see S1 Table for details) when sufficient tissue was available. We assessed cultures for the presence of synaptophysin as a marker for chromaffin cells and GFAP as a marker for sustentacular cells. In addition, we estimated the proportion of proliferating cells based on Ki-67 expression. In general, serum-free medium supported the highest proportion of synaptophysin-positive cells, but no long-term serum-free cultures including these cells were noted, probably because serum-free culture medium leads to the slow depletion of cells. Chromaffin cells visibly fail to prosper in this medium, often typified by poor filopodia development and poor adhesion. Media including some bovine serum, at either 1% or 5%, seemed to achieve a better balance between proliferation of fibroblasts, which may have a supportive function in these cultures, and survival of chromaffin tumour cells. Richer media such as Izal and PC12 led to a predominance of fibroblasts that appeared detrimental to the survival of chromaffin cells, possibly due to the rapid consumption of available metabolites in the media.

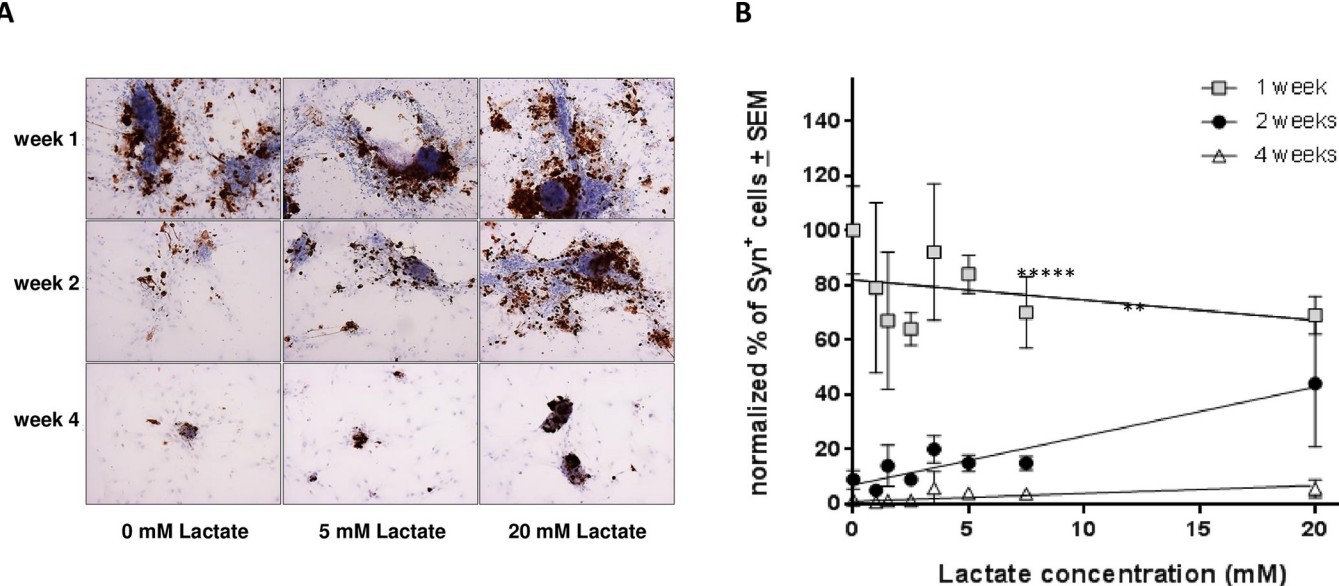

**Fig 7. Lactate supplementation enhances chromaffin cell survival.** A representative experiment (tumour 44) illustrating the beneficial effect of lactate supplementation on the short-term survival of synaptophysin-positive cells of chromaffin morphology. A. Anti-synaptophysin immunohistochemical staining of cultures with 0.0, 5.0 and 20 mM lactate monitored at weeks 1, 2 and 4. B. Regression curves of average scorings of a series of lactate concentrations over the 3 time points (regression analysis, ** $p < 0.01$, ****$p < 0.000001$).

behaviour of human PPGL cultures has ever been published. In addition to the five main culture conditions described here, we also tested other media and FBS concentrations, none of which appeared to significantly promote the survival or proliferation of PPGL tumour cells.

One of the most striking aspects of PPGL tumour culture was the wide variety of morphologies observed in both short- and long-term cultures. Although some data suggested that culture growth characteristics and morphologies were influenced by the specific culture media (primarily FBS levels), different tumours nevertheless often developed distinct characteristics, such as sparse or numerous large or small cell masses, detached round cell masses, macro-fibres of several centimetres in length, or an entirely flattened aspect. Nonetheless, in most cases cultures appeared to be dominated by synaptophysin-negative cells presumably of fibroblast or myofibroblast origin. It remains unclear why these cells display such a wide variety of morphologies and growth characteristics.

## Cell markers

Another goal was to determine which of the well-known chromaffin cell markers commonly employed in the immunohistochemical analysis of formalin-fixed primary PPGLs could also serve as a valid marker in cell cultures. Although antibodies specific for chromogranin and neuron-specific enolase consistently stain chromaffin cells in PPGL FFPE samples, neither of these markers showed consistent staining in PPGL cultures. The best markers proved to be synaptophysin (highly specific, but occasionally also weakly staining connective tissue cells), and tyrosine hydroxylase (although often negative in primary HNPGL tumours and derived cultures). It is worth noting that neither of these markers conclusively confirms the identity of PPGL tumour cells, but based on the widespread staining of early cultures and the general co-occurrence of synaptophysin and tyrosine hydroxylase-positive cells in cultures exhibiting long-term survival of putative tumour cells, together with a consistent cell morphology, we conclude with a considerable degree of confidence that these cells are genuine, persistent

chromaffin tumour cells. These markers can be used to closely monitor tumour cells in future attempts to culture these cells.

## Media

The primary strategy underlying this study was to identify a cell culture containing a cell or cells with potential for proliferation and clonal outgrowth. In practice, we found that five different media formulations together with around 20–25 concurrent tumour cultures was the most feasible number.

Of the five media used long-term, serum-free medium undoubtedly resulted in the purest cultures of synaptophysin-positive chromaffin cells, as this medium was highly effective in suppressing fibroblast growth, even when no special steps such as differential adhesion were taken to purify tumour cells. However, serum-free cultures were often characterized by a sparse and consistently declining number of cells. In general we preferred 1% and 5% FBS medium, as these media allowed a confluent or near-confluent fibroblast cell layer to which synaptophysin-positive cells appeared to be firmly attached. It is possible that other less well-attached tumour cells were gradually lost from culture during media replenishment, despite efforts to avoid this.

The 'Izal' and PC12 media were found to be too "rich" in the sense that fibroblast proliferation and acidification of media were sufficiently rapid to suggest that survival of other cells was compromised.

The data described here will hopefully provide a foundation for current and future efforts to derive human PPGL cell lines. Although our primary goal was not achieved, our findings and the extensive supporting information represent the first detailed description of human PPGL tumour culture to ever appear in literature and as such will hopefully render valuable assistance to others willing to work towards this important goal.

## Lactate

In addition to the various growth factors used in long-term culture, we tested the impact of the putative cancer metabolite, lactate, on chromaffin cell growth and survival in short-term cultures. While lactate did not appear to promote gross tumour cell proliferation as judged by numbers of synaptophysin-positive cells, we found that lactate enhanced the survival of these cells (or, although less likely, the expression of synaptophysin). The exact mechanism behind this effect remains unclear. In addition to its role as a potential direct metabolite [34], lactate may also indirectly support the survival of chromaffin cells via cell signalling. The lactate receptor, GPR81, is known to be involved in neuronal signalling and is thought to play a role in cerebral energy metabolism and the availability of energy substrates [35].

## PPGL models

Over the last two decades, much basic PPGL research effort was focused on unravelling the genetics of these tumours, the first step in understanding any predominantly hereditary cancer. As many research teams are now refocusing on the mechanisms driving tumourigenesis, the need for human and rodent PPGL models has become pressing. In a recent review we discussed the strengths and weaknesses of existing cell model systems relevant to SDHx tumours, including purported human cell lines, and we refer the reader to that publication for a more detailed discussion of the topic [16]. To briefly summarize, the majority of PPGL research to date has been conducted in generic cell lines of diverse and generally non-neuroendocrine origin. Several more specific model systems have been developed but not all are well-characterized. Even the Nf1 mouse-derived MPC/MTT cell lines, the best characterized model system

available [18, 19], have no direct genetic relation to SDH-mutated human tumours. An *sdhb*-KO model developed in zebrafish recapitulates typical phenomena related to loss of SDHB but appears to show no biology relevant to tumour development in humans [36]. The recent development of an SDHB-deficient rat cell line RS0, using a xenograft approach, is more promising [20]. An important aspect of the RS0 study in the current context was the cell culture approach. In preliminary studies cell lines under standard culture conditions survived for around 2 weeks. Survival improved upon culture in 5% $O_2$, but changing to a low- to zero-serum medium together with stem cell-promoting supplements allowed RS0 cells to proliferate as a continuous cell line, appearing as free-floating spheres with a doubling time of approximately 14 days. Even in this case, the rodent origin, together with initial irradiation, the required hypoxic culture conditions and the stem cell factors needed to trigger cell expansion all represent important caveats concerning relevance and/or obstacles to general use. Nevertheless, this study provided interesting and potentially important insights that may be instrumental in increasing the likelihood of the eventual development of cell lines from other species, including humans.

## Human cell lines

Early attempts to culture human paragangliomas include those of Costero and Chevez [21] and Gullotta and Helpap [22]. Little cell proliferation was observed. The most recent report of the culture of exclusively paraganglioma cells was by Arthur Tischler and colleagues in 1981 [23].

Attempts to derive human paraganglioma cell lines include that of Stuschke et al. [37, 38] in the early 1990s. A cell line named EPG1, derived from a metastatic carotid body paraganglioma, was included in radiographic studies and used to generate xenografts in nude mice [39, 40]. However, EPG1 was not characterized using any marker that clearly established cellular identity. Other reports have described human paraganglioma cell lines derived from SDH-mutated jugulotympanic paragangliomas [41, 42]. Again, no meaningful validation of cellular identity was carried out.

Human pheochromocytoma cell lines have also been reported. In 1998, Pfragner and colleagues described the KNA cell line [43], while Venihaki and colleagues independently reported the KAT45 cell line [44]. Both cell lines originated from sporadic pheochromocytomas and, in contrast to the studies described above, both were clearly derived from genuine chromaffin cells, showing a close morphological resemblance to PC12 cells, production of catecholamines or the expression of markers including chromogranin A, human neurofilament protein, S100 and NSE. However, as no follow-up on these cell lines has ever been published, they likely failed to proliferate at some point.

The most recent report of a human pheochromocytoma-derived cell line was by Ghayee and colleagues [45]. Derived from a sporadic adrenal pheochromocytoma and termed hPheo1, this cell line was carefully designated a "progenitor" by the authors (an important and often overlooked caveat) and subsequently immortalized using hTERT. Despite the fact that the authors never claimed that this cell line was derived from the differentiated tumour and expended considerable effort detailing the many inconsistencies between hPheo1 and the original tumour, this cell line is unfortunately still regarded as a tumour cell line by some. No evidence was produced by the authors and none has subsequently emerged (e.g. the ability to generate tumours in nude mice) to suggest that this cell line represents a differentiated pheochromocytoma. The available data suggest that hPheo1 did not originate from a differentiated neuroendocrine tumour cell but from another cell type, possibly of non-neuroendocrine origin. Even the designation "progenitor" is speculative, as nothing concrete is known about pheochromocytoma progenitor cells.

As the data presented in the present study and this brief overview of previous efforts attests, a human tumour-derived paraganglioma-pheochromocytoma cell line remains an important challenge.

## Future directions

Often considered 'difficult', culture of human SDHx-related paraganglioma and pheochromocytoma tissue has been neglected (with notable exceptions) and the few studies undertaken often remain unpublished, frustrating potential advance. As paraganglioma/pheochromocytoma research transitions from genetics to the study of the mechanisms underlying tumorigenesis, the current dearth of models will become ever more problematic.

Approaches such as Tet-On-regulated oncogene-mediated immortalization, which allows controlled cell proliferation, have been insufficiently explored in paraganglioma/pheochromocytoma. It is worth recalling that head and neck paragangliomas show an in vivo doubling time of four years [11] and malignant tumours have an 85% 5-year survival rate [46]. This low level of proliferation suggests that even cells from malignant tumours may divide too slowly in culture to be of practical value, leaving regulated oncogene-mediated immortalization as the only viable option. Other interesting alternative approaches include patient-derived tumour xenograft models, patient-derived tumour organoid models, as well as iPSCs combined with CRISPR/Cas as a model for chromaffin-derived tumours. Whether these options will be actively pursued remains to be seen.

## Conclusion

In the current study we showed for the first time that standard approaches to the culture of human PPGL, however varied and persistent, are inadequate to achieve cell proliferation of human chromaffin tumour cells, possibly reflecting the indolent nature of primary PPGLs and even their metastases. Nonetheless, we also found convincing evidence that a minority of these cells survive for long periods and as such may provide a basis from which to explore more innovative methods of cell culture. As we have previously argued [16], stem-cell and organoid culture techniques have been insufficiently explored (although recent verbal reports have been disappointing), and perhaps most importantly, techniques of reversible oncogene-driven proliferation deserve more attention.

## Supporting information

**S1 Fig. Overview of immunocytochemical staining for protein markers in control cell lines PC-12, SH-SY5Y and VH10hTneo.**
(PDF)

**S2 Fig. Analysis of tumour explant culture to determine cellularity.**
(PDF)

**S3 Fig. Antibody stainings of formalin-fixed paraffin-embedded (FFPE) PPGL tumors.**
(PDF)

**S4 Fig. The digestion and culture of human paragangliomas-pheochromocytomas.**
(PDF)

**S1 Table. Media, supplements, antibodies and immunocytochemical scoring method used in the study.**
(XLSX)

**S2 Table. Immunocytochemical scoring of control cell lines and a range of PGL tumour cultures based on morphological criteria.**
(XLSX)

**S3 Table. Estimated (not scored) percentages of Immunocytochemical-positive cells in primary PGL cultures.**
(XLSX)

## Acknowledgments

We acknowledge the contribution of Ing. Caro Meijer to the maintenance of paraganglioma-pheochromocytoma cell cultures.

## Author Contributions

**Conceptualization:** Jean-Pierre Bayley, Peter Devilee.

**Data curation:** Jean-Pierre Bayley, Heggert G. Rebel, Juan Zhang.

**Formal analysis:** Jean-Pierre Bayley, Heggert G. Rebel, Juan Zhang.

**Funding acquisition:** Jean-Pierre Bayley.

**Investigation:** Jean-Pierre Bayley, Heggert G. Rebel, Kimberly Scheurwater, Dominique Duesman, Juan Zhang.

**Methodology:** Jean-Pierre Bayley, Heggert G. Rebel, Juan Zhang, Peter Devilee.

**Project administration:** Jean-Pierre Bayley, Peter Devilee.

**Resources:** Francesca Schiavi, Esther Korpershoek, Jeroen C. Jansen, Abbey Schepers.

**Supervision:** Jean-Pierre Bayley, Peter Devilee.

**Writing – review & editing:** Peter Devilee.

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
