## [Decision Letter · Decision Letter 0]

4 Jul 2022

PONE-D-22-14862Long-term in vitro 2D-culture of SDHB and SDHD-related human paragangliomas and pheochromocytomasPLOS ONE

Dear Dr. Bayley,

Thank you for submitting your manuscript to PLOS ONE. After careful consideration, we feel that it has merit but does not fully meet PLOS ONE’s publication criteria as it currently stands. Therefore, we invite you to submit a revised version of the manuscript that addresses the points raised during the review process.

 This is a very interesting topic of HN-PGL cultivation. However, it is necessary to answer some reviewers' questions. And also, to fix some of the bugs described by reviewer 1.

We look forward to receiving your revised manuscript.

Kind regards,

Ales Vicha, M.D., PhD

Academic Editor

PLOS ONE

Journal Requirements:

Additional Editor Comments:

This is a very interesting topic of HN-PGL cultivation. However, it is necessary to answer some reviewers' questions. And also, to fix some of the bugs described by reviewer 1.

Reviewers' comments:

Reviewer's Responses to Questions

**Comments to the Author**

1. Is the manuscript technically sound, and do the data support the conclusions?

Reviewer #1: Yes

Reviewer #2: Yes

2. Has the statistical analysis been performed appropriately and rigorously? 

Reviewer #1: No

Reviewer #2: Yes

3. Have the authors made all data underlying the findings in their manuscript fully available?

Reviewer #1: Yes

Reviewer #2: Yes

4. Is the manuscript presented in an intelligible fashion and written in standard English?

Reviewer #1: Yes

Reviewer #2: Yes

5. Review Comments to the Author

Reviewer #1: The authors report an an impressive number of attempts to produce paraganglioma/pheochromocytoma cell line from human tumor samples. This effort is commendable, as few such cell lines are available at the moment. Unfortunately, they did not really succeed in their efforts. Still, it can be of interest to people attempting the same. However, the manuscript is somewhat descriptive and it would help to provide better analysis of the studied tumor samples.

1. The authors mention SDHB and SDHD in the title. Could they provide statistics of the survival in cell culture based on SDH genotype? Where there any differences between samples harboring SDH mutations and and other samples?

2. The authors should discuss the work by Ghayee et al (PMID 23785438) that actually describes a human pheochromocytoma-derived cell line. Why did the authors not consider tert-immortalization?

3. It would be useful to indicate that SDHA-D are all subunits of the same respiratory complex

There are some typos/mistakes:

153 is it 20g od 200g?

297 XX tumors?

298 table s1? There is no table 1.

438 ref Warburg?

Reviewer #2: Paraganglioma (PPGL) and pheochromocytoma (PHEO) are relatively rare endocrine tumours, whose research is hindered by existence of virtually no cell line that could be used in vitro. In this manuscript the authors attempt to grow and characterize a number of cells from PPGL patients. This is a vast and extensive study that is of interest, albeit it seems that in the end, there is no palpable result such as possibility to provide a viable PPGL cell line that could be used by researchers for long-term tissue culture. Notwithstanding this, the research has merit in giving some grounds as how to approach the problem of deriving a PPGL cell line, and importantly, too, with mutations in SDH subunits.

The work needs to be revised slightly. For example, the authors do not refer to the cell line hPheo1 that is derived from a PHEO patients and is available for use in tissue culture. While it would have been nice to use this cell line in this work as a 'control' cell line, the authors should at least mention it in the Introduction and/or in the Discussion. I feel that there should be more in the concluding parts of the Discussion as to where to go from now and how this work contributes to the topic. Have the authors thought about PPGL PDX models and perhaps deriving a cell line or two from such a setting? Perhaps this is something also to add in the Discussion.

6. PLOS authors have the option to publish the peer review history of their article (what does this mean?). If published, this will include your full peer review and any attached files.

Reviewer #1: No

Reviewer #2: No

---

## [Author Response · Author response to Decision Letter 0]

16 Aug 2022

Response to Reviewers PONE-D-22-14862

Long-term in vitro 2D-culture of SDHB and SDHD-related human paragangliomas and pheochromocytomas

PLOS ONE

Reviewer's Responses to Questions

Comments to the Author

1. Is the manuscript technically sound, and do the data support the conclusions?

Reviewer #1: Yes

Reviewer #2: Yes

2. Has the statistical analysis been performed appropriately and rigorously? 

Reviewer #1: No

Reviewer #2: Yes

A: We answer reviewer 1 in point 1 below. 

3. Have the authors made all data underlying the findings in their manuscript fully available?

Reviewer #1: Yes

Reviewer #2: Yes

4. Is the manuscript presented in an intelligible fashion and written in standard English?

Reviewer #1: Yes

Reviewer #2: Yes

5. Review Comments to the Author

Reviewer #1: The authors report an impressive number of attempts to produce paraganglioma/pheochromocytoma cell line from human tumor samples. This effort is commendable, as few such cell lines are available at the moment. Unfortunately, they did not really succeed in their efforts. Still, it can be of interest to people attempting the same. However, the manuscript is somewhat descriptive and it would help to provide better analysis of the studied tumor samples.

1. The authors mention SDHB and SDHD in the title. Could they provide statistics of the survival in cell culture based on SDH genotype? Where there any differences between samples harboring SDH mutations and and other samples?

A: Unfortunately this is not an analysis that we carried out. It would also be technically very challenging due to the need to repeatedly sample a culture and subculture on chamber slides in order to confirm cellular identity using immunohistochemistry. Without a clear rationale at the moment to support the idea that variants in SDHD or SDHB might differentially promote cell survival, I think this experiment is one for the future. We first need to identify basic conditions under which SDH tumors survive and proliferate.

2. The authors should discuss the work by Ghayee et al (PMID 23785438) that actually describes a human pheochromocytoma-derived cell line. Why did the authors not consider tert-immortalization?

A: We are aware of the work of Ghayee and colleagues, as well as the efforts of many others. We have discussed this work in detail in a recent review (Advances in paraganglioma-pheochromocytoma cell lines and xenografts. Bayley JP, Devilee P. Endocr Relat Cancer. 2020 Dec;27(12):R433-R450. doi: 10.1530/ERC-19-0434). The hPheo1 cell line was described as a “progenitor” cell line (Ghayee et al. 2013) and a detailed reading of the paper does not indicate that this cell line is actually derived from pheochromocytoma tumour cells. Various lines of evidence presented by the authors clearly indicate that it is not in fact a tumour-derived cell line. We have now added a paragraph discussing the efforts of Ghayee and others regarding human cell lines (line 569-603).

Regarding hTERT, we have noted the efforts of Ghayee et al. and have heard of other unsuccessful uses of hTERT via personal communications. Nevertheless, immortalization is an attractive route and perhaps the only viable option at this juncture (see our Review mentioned above). We therefore carried out a series of experiments using cMyc as the immortalization agent, with very promising initial results (manuscript in preparation).

3. It would be useful to indicate that SDHA-D are all subunits of the same respiratory complex.

A: A sentence has been added to this effect (line 51).

There are some typos/mistakes:

153 is it 20g od 200g?

A: 20g is correct. A sentence has been added to explain this more clearly.

297 XX tumors?

A: Has now been corrected.

298 table s1? There is no table 1.

A: There is now.

438 ref Warburg?

A: This has now been corrected.

Reviewer #2: Paraganglioma (PPGL) and pheochromocytoma (PHEO) are relatively rare endocrine tumours, whose research is hindered by existence of virtually no cell line that could be used in vitro. In this manuscript the authors attempt to grow and characterize a number of cells from PPGL patients. This is a vast and extensive study that is of interest, albeit it seems that in the end, there is no palpable result such as possibility to provide a viable PPGL cell line that could be used by researchers for long-term tissue culture. Notwithstanding this, the research has merit in giving some grounds as how to approach the problem of deriving a PPGL cell line, and importantly, too, with mutations in SDH subunits.

The work needs to be revised slightly. For example, the authors do not refer to the cell line hPheo1 that is derived from a PHEO patients and is available for use in tissue culture. While it would have been nice to use this cell line in this work as a 'control' cell line, the authors should at least mention it in the Introduction and/or in the Discussion. I feel that there should be more in the concluding parts of the Discussion as to where to go from now and how this work contributes to the topic. Have the authors thought about PPGL PDX models and perhaps deriving a cell line or two from such a setting? Perhaps this is something also to add in the Discussion.

A: We have now added a paragraph discussing hPheo1 and other human cell line efforts. As we discuss here, and more extensively in our earlier review (Advances in paraganglioma-pheochromocytoma cell lines and xenografts. Bayley JP, Devilee P. Endocr Relat Cancer. 2020 Dec;27(12):R433-R450. doi: 10.1530/ERC-19-0434), we don’t agree that hPheo1 is a pheochromocytoma tumour cell line. Ghayee and colleagues also never made that claim and no evidence has since emerged to suggest that they were incorrect. 

As for discussion of future directions, we were initially reluctant to make this already long article any longer, especially as we have recently published an 8000 word review on this subject. In hindsight we agree with the reviewer that some discussion of the topic is necessary and have now added an additional paragraph (line 570-603).

6. PLOS authors have the option to publish the peer review history of their article (what does this mean?). If published, this will include your full peer review and any attached files.

Do you want your identity to be public for this peer review? For information about this choice, including consent withdrawal, please see our Privacy Policy.

Reviewer #1: No

Reviewer #2: No

---

## [Decision Letter · Decision Letter 1]

30 Aug 2022

Long-term in vitro 2D-culture of SDHB and SDHD-related human paragangliomas and pheochromocytomas

PONE-D-22-14862R1

Dear Dr. Bayley,

We’re pleased to inform you that your manuscript has been judged scientifically suitable for publication and will be formally accepted for publication once it meets all outstanding technical requirements.

Kind regards,

Ales Vicha, M.D., PhD

Academic Editor

PLOS ONE

Additional Editor Comments (optional):

Reviewers' comments:

Reviewer's Responses to Questions

**Comments to the Author**

1. If the authors have adequately addressed your comments raised in a previous round of review and you feel that this manuscript is now acceptable for publication, you may indicate that here to bypass the “Comments to the Author” section, enter your conflict of interest statement in the “Confidential to Editor” section, and submit your "Accept" recommendation.

Reviewer #1: All comments have been addressed

Reviewer #2: All comments have been addressed

2. Is the manuscript technically sound, and do the data support the conclusions?

Reviewer #1: Yes

Reviewer #2: Yes

3. Has the statistical analysis been performed appropriately and rigorously? 

Reviewer #1: N/A

Reviewer #2: Yes

4. Have the authors made all data underlying the findings in their manuscript fully available?

Reviewer #1: Yes

Reviewer #2: Yes

5. Is the manuscript presented in an intelligible fashion and written in standard English?

Reviewer #1: Yes

Reviewer #2: Yes

6. Review Comments to the Author

Reviewer #1: The authors satisfactorily addressed/explained the issues raised. It is nice that they also provide some future directions. It is of course pity that the statistical analysis could not be performed. Furthermore, they seem rather combative towards the Ghayee paper, and the Pheo-1 cell line, and now they definitely acknowledge its existence. Perhaps this will simulate discussion in the community. For author's information: While not being linked to the Ghayee paper in any way, we used PHENO-1 cell line and it produces tumors in NSG mice that are similar morphologically to human tumors (as assessed by a trained human pathologist that routinely scores pheochromocytoma/paragangliomas). Also personal communication with other investigators confirms this. Hence, the authors might consider moderating their statements re: Pheo-1 slightly.

Reviewer #2: The authors dealt appropriately with the points raised during revision, so that the manuscript can now be accepted.

7. PLOS authors have the option to publish the peer review history of their article (what does this mean?). If published, this will include your full peer review and any attached files.

Reviewer #1: No

Reviewer #2: No

---

## [Editor Report · Acceptance letter]

2 Sep 2022

PONE-D-22-14862R1 

Long-term in vitro 2D-culture of SDHB and SDHD-related human paragangliomas and pheochromocytomas 

Dear Dr. Bayley:

I'm pleased to inform you that your manuscript has been deemed suitable for publication in PLOS ONE. Congratulations! Your manuscript is now with our production department. 

Kind regards, 

on behalf of

Dr. Ales Vicha 

Academic Editor

PLOS ONE